ecology

stable isotopes, $\delta^{13}$C and $\delta^{15}$N, cuttlefish, cephalopod, natural tracers

**Author for correspondence:**
R. J. David Wells
e-mail: wellsr@tamug.edu

Electronic supplementary material is available online https://doi.org/10.6084/m9.figshare.c.5577714.

# Regional patterns of $\delta^{13}$C and $\delta^{15}$N for European common cuttlefish (*Sepia officinalis*) throughout the Northeast Atlantic Ocean and Mediterranean Sea

R. J. David Wells[1,2], Jay R. Rooker[1,2], Piero Addis[3], Haritz Arrizabalaga[4], Miguel Baptista[5], Giovanni Bearzi[6], Igaratza Fraile[4], Thomas Lacoue-Labarthe[7], Emily N. Meese[1], Persefoni Megalofonou[8], Rui Rosa[5], Ignacio Sobrino[9], Antonio V. Sykes[10] and Roger Villanueva[11]

[1]Department of Marine Biology, Texas A&M University, 1001 Texas Clipper Road, Galveston, TX 77553, USA
[2]Department of Ecology and Conservation Biology, Texas A&M University, College Station, TX 77843, USA
[3]Department of Environmental and Life Science, University of Cagliari, Via Fiorelli 1, 09126 Cagliari, Italy
[4]AZTI, Marine Research, Basque Research and Technology Alliance (BRTA), Herrera Kaia-Portualdea z/g, 20110 Pasaia – Gipuzkoa, Spain
[5]Marine and Environmental Sciences Centre, Laboratorio Maritimo da Guia, Faculdade de Ciencias, Universidade de Lisboa, Avenida Nossa Senhora do Cabo, 939, 2750-374 Cascais, Portugal
[6]Dolphin Biology and Conservation, Cordenons, Pordenone, Italy
[7]LIttoral Environnement et Sociétés (LIENSs) – UMR 7266 Bâtiment ILE, 2, rue Olympe de Gouges, 17000 La Rochelle, France
[8]Department of Biology, National and Kapodistrian University of Athens, 15784 Athens, Greece
[9]Instituto Español de Oceanografía, Puerto Pesquero s/n, 11006, Cádiz, Spain
[10]Center of Marine Sciences, Universidade do Algarve Campus de Gambelas, 8005-139 Faro, Portugal
[11]Institut de Ciències del Mar (CSIC), Passeig Maritim, 37-49, 08003, Barcelona, Spain

RJDW, 0000-0002-1306-0614; JRR, 0000-0002-5934-7688

Stable isotope compositions of carbon and nitrogen (expressed as $\delta^{13}$C and $\delta^{15}$N) from the European common cuttlefish

(*Sepia officinalis*) were measured in order to evaluate the utility of using these natural tracers throughout the Northeast Atlantic Ocean and Mediterranean Sea (NEAO-MS). Mantle tissue was obtained from *S. officinalis* collected from 11 sampling locations spanning a wide geographical coverage in the NEAO-MS. Significant differences of both $\delta^{13}C$ and $\delta^{15}N$ values were found among *S. officinalis* samples relative to sampling location. $\delta^{13}C$ values did not show any discernable spatial trends; however, a distinct pattern of lower $\delta^{15}N$ values in the Mediterranean Sea relative to the NEAO existed. Mean $\delta^{15}N$ values of *S. officinalis* in the Mediterranean Sea averaged 2.5‰ lower than conspecifics collected in the NEAO and showed a decreasing eastward trend within the Mediterranean Sea with the lowest values in the most eastern sampling locations. Results suggest $\delta^{15}N$ may serve as a useful natural tracer for studies on the population structure of *S. officinalis* as well as other marine organisms throughout the NEAO-MS.

# 1. Introduction

The European common cuttlefish, *Sepia officinalis* Linnaeus, 1758 is a coastal nektobenthic species ranging from the Shetland Islands through the Northeast Atlantic Ocean and Northwest Africa into the Mediterranean Sea [1]. This species constitutes one of the most economically valuable cephalopod resources in the Northeast Atlantic Ocean, supporting an important fishery resource [2,3]. *Sepia officinalis* has a relatively short lifespan of 1–2 years, early sexual maturity and an extended spawning season laying eggs on the seafloor with direct benthic, large hatchlings [4,5]. Given this species geographical distribution combined with limited dispersal, it has been a targeted model species to examine connectivity throughout the Northeast Atlantic Ocean and Mediterranean Sea (hereafter NEAO-MS) [6].

Natural biomarkers such as stable isotopes are commonly used to examine food web structure and ecosystem connectivity in marine environments [7,8]. Stable isotopes of carbon ($\delta^{13}C$) and nitrogen ($\delta^{15}N$) are particularly useful tracers due to their natural abundance being influenced by the environment and ease of measurement in body tissues without having to track individuals in a population. $\delta^{13}C$ is traditionally used to trace carbon pathways because little fractionation occurs between predator and prey, and different primary producers (energy sources) often have unique $\delta^{13}C$ values [9]. $\delta^{13}C$ values of consumers are a product of the primary producers' composition and influenced by the dissolved inorganic carbon (DIC) pool, as well as local abiotic factors including sea surface temperature, and can differ across ocean basins [10] and region-specific freshwater to marine gradients [9]. $\delta^{15}N$ becomes enriched with increasing trophic level and is used to infer trophic position [7], but can also differ at the base of the food web. Depending upon the types of nutrients available to stimulate growth, $\delta^{15}N$ values can be used to track energy flow in high-nutrient (nitrate) and low-nutrient ($N_2$ fixation) ecosystems as well as new nitrogen (upwelled nitrate) versus regenerated nitrogen (ammonia, urea). Combining both $\delta^{13}C$ and $\delta^{15}N$ offers the potential to study the connectivity and population structure of species because longitudinal and latitudinal gradients exist throughout marine ecosystems [11,12], including the NEAO-MS [8,13].

The population structure of *S. officinalis* has been found throughout the NEAO-MS [14]. Rooker *et al.* [6] used stable isotopes of $\delta^{13}C$ and $\delta^{18}O$ in the cuttlebones and found regional differences throughout the NEAO-MS due to different environmental conditions among regions, in addition to potential movement and connectivity between the Strait of Gibraltar and western Mediterranean Sea. Another study combined genetics, morphometrics and trace elements in cuttlebones with results suggesting discrete population structure within the eastern Mediterranean Sea [15]. Drábková *et al.* [16] showed the degree of genetic diversification of *S. officinalis* throughout the Mediterranean Sea and suggested this pattern resulted from a low dispersion capability and a complex structure with different degrees of genetic exchange among these subpopulations. Further, Pasqual *et al.* [17] performed a meta-analysis of species connectivity (based on life-history traits) throughout the Mediterranean Sea supporting limited connectivity and a high probability of population structuring for species such as *S. officinalis*. Previous studies have found patterns of $\delta^{13}C$ and $\delta^{15}N$ in other marine species throughout the NEAO-MS suggesting the potential to use these tracers to evaluate population structure and connectivity [13,18].

The primary purpose of this study was to examine patterns in stable isotopes of $\delta^{13}C$ and $\delta^{15}N$ from mantle tissue of *S. officinalis* collected throughout the NEAO-MS in order to evaluate the utility of using these natural tracers as tools to examine population structure. Previous studies on *S. officinalis* have used

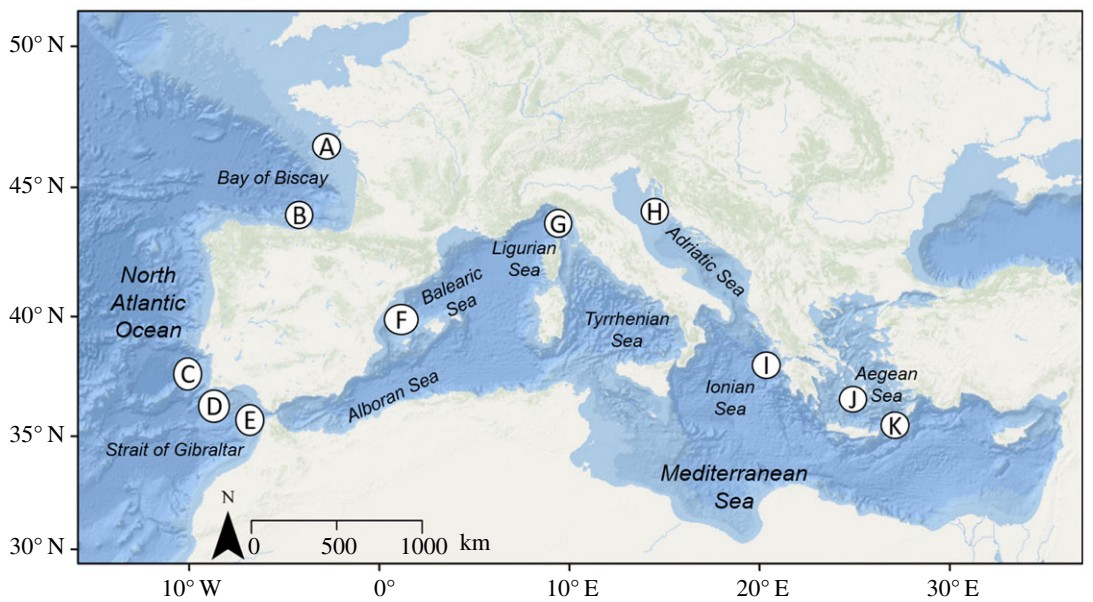

**Figure 1.** Map of 11 sampling locations for *Sepia officinalis* from the Northeast Atlantic Ocean and Mediterranean Sea (NEAO-MS). Sample locations include Bay of Biscay–North (A), Bay of Biscay–South (B), Portugal–West (C), Portugal–South (D), Gulf of Cadiz (E), Balearic Sea (F), Ligurian Sea (G), Adriatic Sea (H), Gulf of Corinth (I), Aegean Sea–West (J) and Aegean Sea–East (K).

a variety of techniques including genetics [19], cuttlebone geochemistry [6,20] and electronic tagging [4], thus this study will expand upon the ecological toolbox available to researchers.

# 2. Material and methods

Samples of *S. officinalis* were collected over a 1-year period (2013–2014) from 11 locations throughout the NEAO-MS (figure 1). Sample locations included five locations within the NEAO: Bay of Biscay–North (A), Bay of Biscay–South (B), Portugal–West (C), Portugal–South (D) and Gulf of Cadiz (E). Six locations within the Mediterranean Sea included Balearic Sea (F), Ligurian Sea (G), Adriatic Sea (H), Gulf of Corinth (I), Aegean Sea–West (J) and Aegean Sea–East (K). Samples were acquired from artisanal or commercial trawlers, and local market sampling. For market sampling, special care was taken to acquire small body sizes by local fishers in order to ensure samples were from the specific region of collection. No special permission to conduct the research was needed and no ethical assessment was required. Upon collection, *S. officinalis* samples were measured using straight mantle length (ML) to the nearest mm and white muscle tissue samples were excised from the mantle and immediately freeze dried or preserved in 70% ethanol and later freeze dried. Subsamples of cuttlefish were paired with both (i) immediately freeze dried and (ii) ethanol-preserved specimens later freeze dried from Bay of Biscay–South ($n = 20$) and Gulf of Cadiz ($n = 19$) sample locations in order to assess if sample preservation techniques affected $\delta^{13}$C and $\delta^{15}$N.

Freeze-dried muscle tissue for stable isotope analysis was homogenized using a mortar and pestle, weighed, wrapped in tin capsules and shipped to the University of California at Davis' Stable Isotope Facility for analysis. Analysis of muscle tissue $\delta^{13}$C and $\delta^{15}$N was carried out using an elemental analyser (PDZ Europa ANCA-GSL) coupled with an isotope ratio mass spectrometer (PDZ Europa 20–20). The long-term standard deviation of UC Davis' stable isotope facility is 0.2‰ for $\delta^{13}$C and 0.3‰ for $\delta^{15}$N. Lipids were not extracted from tissue; however, C:N ratios were low (less than or equal to 3.75) across the size spectrum of *S. officinalis* samples, indicating a low lipid content and little influence of lipids on tissue $\delta^{13}$C values [21]. Mean C:N ratio was $3.21 \pm 0.13$ s.d. and ranged from 2.87 to 3.75. A study by Ruiz-Cooley *et al.* [22] also showed a significant effect on stable isotope values as a result of lipid extractions on squid tissues and recommended researchers use caution with this approach for ecological studies on cephalopods. Isotopic ratios are reported relative to Vienna PeeDee belemnite for carbon and atmospheric $N_2$ for nitrogen.

Multivariate analysis of covariance (MANCOVA) was used to test for differences in muscle tissue $\delta^{13}$C and $\delta^{15}$N. Dependent variables included both $\delta^{13}$C and $\delta^{15}$N, cuttlefish size (ML) was used as

**Table 1.** Summary statistics of cuttlefish collected from 11 locations throughout the NEAO-MS. Mean $\delta^{13}$C and $\delta^{15}$N ($\pm 1$ s.d.), sample size ($n$), mean size (mm) ($\pm 1$ s.d.) and size range (mm).

| sample location | $\delta^{13}$C | $\delta^{15}$N | $n$ | mean size | size range |
|---|---|---|---|---|---|
| Bay of Biscay–North | −16.77 (0.36) | 14.21 (0.52) | 20 | 144.0 (9.4) | 130–170 |
| Bay of Biscay–South | −17.63 (0.26) | 12.84 (0.39) | 64 | 120.4 (10.9) | 100–152 |
| Portugal–West | −15.61 (0.89) | 11.98 (0.77) | 23 | 79.9 (37.7) | 18–153 |
| Portugal–South | −13.29 (1.16) | 12.22 (0.45) | 20 | 79.5 (15.9) | 55–120 |
| Gulf of Cadiz | −16.71 (0.50) | 12.40 (0.88) | 20 | 108.2 (20.1) | 73–144 |
| Balearic Sea | −16.58 (0.24) | 10.13 (0.29) | 20 | 92.3 (7.9) | 79–107 |
| Ligurian Sea | −16.33 (0.44) | 10.93 (0.68) | 19 | 58.7 (7.6) | 45–70 |
| Adriatic Sea | −16.26 (0.58) | 10.83 (0.72) | 17 | 52.1 (5.0) | 45–60 |
| Gulf of Corinth | −16.01 (0.78) | 9.78 (0.89) | 19 | 103.4 (16.9) | 77–127 |
| Aegean Sea–West | −14.68 (2.60) | 9.34 (0.75) | 8 | 95.1 (14.8) | 75–122 |
| Aegean Sea–East | −16.93 (0.66) | 9.34 (0.69) | 6 | 98.0 (12.9) | 82–116 |

the covariate, and sampling location as the independent factor. Pillai's trace statistic was used to test for significance. Univariate tests for both $\delta^{13}$C and $\delta^{15}$N were also performed using analysis of covariance (ANCOVA) and *a posteriori* differences among means were analysed with Tukey's honestly significant difference (HSD) test. A *t*-test was used to test for differences in $\delta^{13}$C and $\delta^{15}$N of cuttlefish tissue based on preservation technique (freeze-dried versus ethanol-preserved specimens). Normality was evaluated using a Shapiro–Wilk test and the equal variance assumption was assessed by the Spearman rank correlation between the absolute value of the residuals and the observed value of the dependent variable. Quadratic discriminant function analysis (QDFA) was used to evaluate the best possible classification accuracy of $\delta^{13}$C and $\delta^{15}$N values of cuttlefish tissue based on (i) sampling location (sites A–K) and (ii) NEAO and MS regions. Statistical significance for all tests was determined at the alpha level of 0.05.

## 3. Results

A total of 236 individual *S. officinalis* tissue samples were analysed for $\delta^{13}$C and $\delta^{15}$N (table 1). Overall mean size of *S. officinalis* was 99 mm ML ($\pm 31$ s.d.) ranging from 18 to 170 mm. Comparisons between preservation techniques on $\delta^{13}$C and $\delta^{15}$N values of *S. officinalis* tissue were negligible supporting the use of either ethanol or freeze-dried samples. Subsamples of *S. officinalis* from the Bay of Biscay–South ($n = 20$) using both preservation methods had no significant effect on $\delta^{13}$C ($p = 0.481$) or $\delta^{15}$N ($p = 0.223$). Similarly, subsamples from the Gulf of Cadiz ($n = 19$) had similar $\delta^{13}$C ($p = 0.128$) and $\delta^{15}$N ($p = 0.596$) values using both preservation techniques (electronic supplementary material, data).

A significant main effect of sampling location was found for both $\delta^{13}$C and $\delta^{15}$N (MANCOVA, $p < 0.001$). Significant differences in $\delta^{13}$C values of *S. officinalis* were found among sampling locations (ANCOVA, $p < 0.001$) with no main effect of cuttlefish size (ANCOVA, $p = 0.900$). *Sepia officinalis* collected from Portugal–South had significantly higher $\delta^{13}$C values than samples collected from all other sampling locations (Tukey HSD, $p < 0.001$) (figure 2*a* and table 1). Mean $\delta^{13}$C values from Portugal–South (−13.29‰ $\pm 1.16$) averaged 1.39–4.34‰ higher than $\delta^{13}$C values from the other sampling locations (figure 3 and table 1). Lowest $\delta^{13}$C values were found in *S. officinalis* tissues collected from Bay of Biscay–South (mean −17.63‰ $\pm 0.26$) and significantly differed from all other sampling locations with the exception of three locations in the Mediterranean Sea (Ligurian Sea, Adriatic Sea, Aegean Sea–East) (figure 3).

Mean $\delta^{15}$N values of *S. officinalis* tissue differed significantly among sampling locations (ANCOVA, $p < 0.001$) with higher values in the NEAO relative to the Mediterranean Sea (figure 2*b*). A general pattern of higher $\delta^{15}$N values of *S. officinalis* collected in the NEAO with a gradual depletion into the Mediterranean Sea existed with a distinct gradient around the Strait of Gibraltar (figure 2*b*). In general, mean $\delta^{15}$N values of *S. officinalis* in the Mediterranean Sea averaged 2.51‰ lower than conspecifics collected in the NEAO and continued to decrease eastward within the Mediterranean Sea

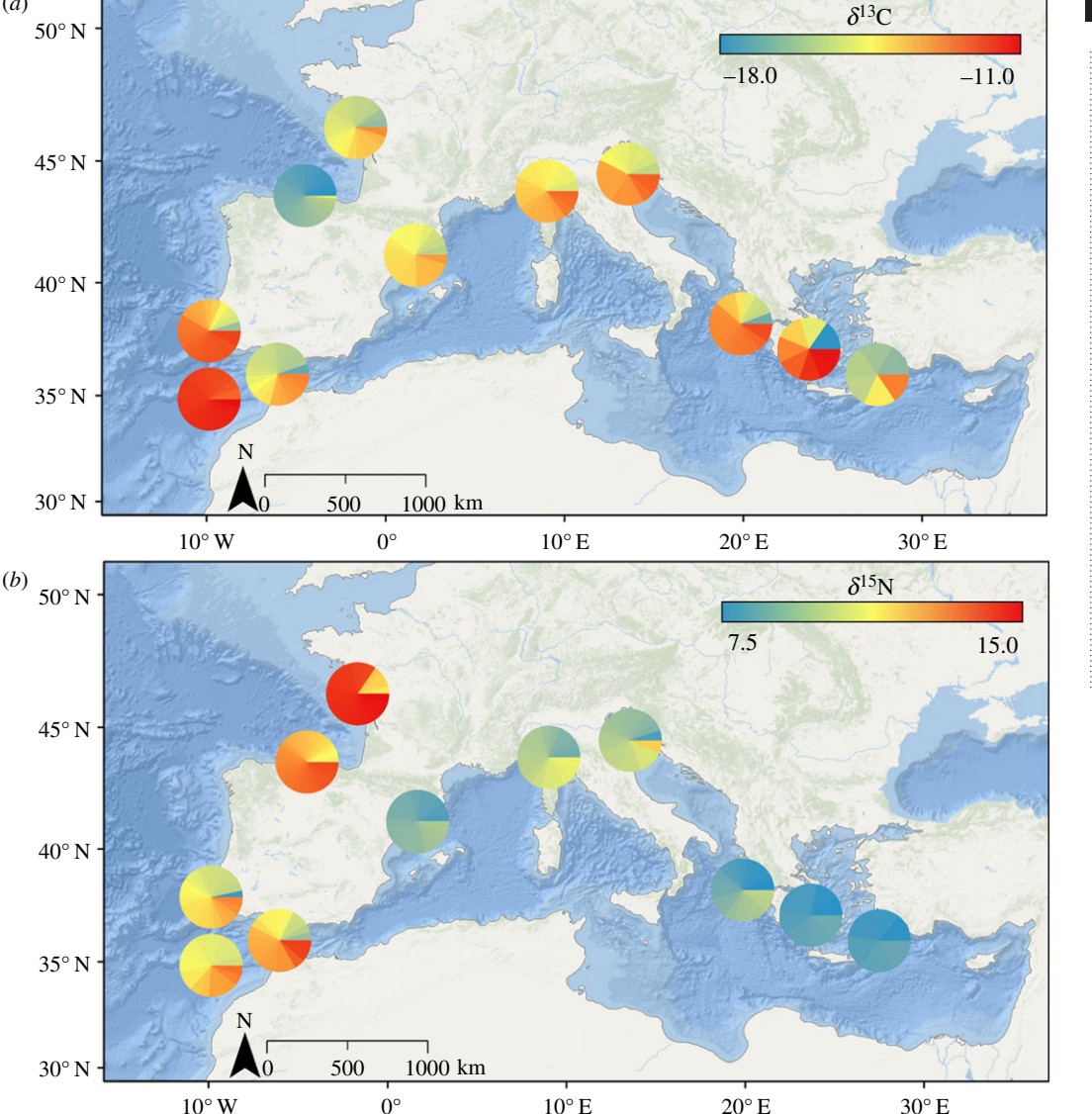

**Figure 2.** Spatial visualizations of $\delta^{13}$C (*a*) and $\delta^{15}$N (*b*) from muscle tissue of *Sepia officinalis* were collected throughout the Northeast Atlantic Ocean and Mediterranean Sea (NEAO-MS). The pie charts at each sampling location display individual variability, with each wedge representing an individual *S. officinalis* sampled at that location.

with lowest values in the most eastern sampling locations (Gulf of Corinth, Aegean Sea–West, Aegean Sea–East) (figure 3). While size ranges of cuttlefish slightly differed among sample locations, no main effect of size was found on $\delta^{15}$N (ANCOVA, $p = 0.481$).

Results from QDFA showed moderate overall classification success of 60% among the 11 sampling locations. Classification success was highest at sampling locations in the Atlantic with Bay of Biscay–North at 90% and Bay of Biscay–South at 98%, while lowest classification success was from sampling locations in the Mediterranean Sea (0% success for both Adriatic Sea and Aegean Sea–East). Overall classification success was 93% when contrasting between the Atlantic (96%) and Mediterranean Sea (89%) highlighting the differences in $\delta^{13}$C and $\delta^{15}$N of cuttlefish tissue from the two regions.

## 4. Discussion

Life-history strategy of *S. officinalis* combined with previous tracer studies suggests limited dispersal and connectivity leading to population structuring in the NEAO-MS. A relatively short lifespan, early sexual maturity and a nektobenthic strategy lacking a pelagic larval stage supports population structuring of this species throughout its range [17,23]. Additionally, genetic studies of *S. officinalis* within the

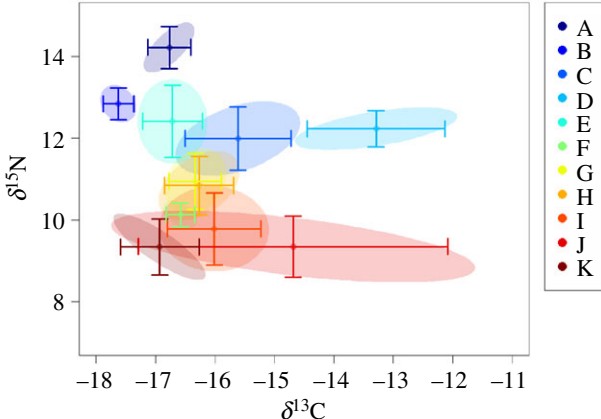

**Figure 3.** Biplot of mean $\delta^{13}C$ and $\delta^{15}N$ values (centre of ellipses) with ± 1 s.d. (bars) and 50% confidence intervals (shaded ellipses) for *Sepia officinalis* muscle tissues collected from each of the 11 sampling locations (A to K).

NEAO-MS highlight the complex population structure and a high degree of genetic diversification at a relatively fine spatial scale [14,16,24]. Pérez-Losada *et al.* [19] suggested that the population structure of *S. officinalis* in the NEAO-MS was driven by isolation through distance and the results of stable isotopes in muscle tissue presented here show support for this based on the isoscapes throughout the range examined. These results parallel those from a similar study conducted by Rooker *et al.* [6] that found area-specific geochemical signatures in cuttlebone chemistry over the same geographical range. While this study did not have the spatial resolution to examine potential connectivity through the Strait of Gibraltar as Rooker *et al.* [6], results show clear gradients and potential for stable isotopes of $\delta^{13}C$ and $\delta^{15}N$ as natural tracers for this species.

Spatial patterns of $\delta^{15}N$ in *S. officinalis* muscle tissue showed a distinct gradient with higher values in samples collected from the NEAO and a gradual depletion into the Mediterranean Sea from west to east. These findings parallel those of McMahon *et al.* [8] where authors performed a meta-analysis of $\delta^{15}N$ for zooplankton examining spatial patterns throughout the Atlantic Ocean and found $\delta^{15}N$ values were on average 2‰ higher than values in the Mediterranean Sea. McMahon *et al.* [8] also found a similar gradient within the Mediterranean Sea of higher $\delta^{15}N$ values in the west near the Strait of Gibraltar relative to the east. Similar $\delta^{15}N$ patterns have been found for a wide variety of marine taxa ranging from krill [25], fishes [18,26], whales [13] and seabirds [27] with consistently higher values in the NEAO relative to the Mediterranean Sea. Patterns of $\delta^{15}N$ are most likely due to the differential utilization of nitrogen sources. Higher latitudes of the Atlantic are fuelled by nitrate in contrast with the oligotrophic Mediterranean Sea where nitrogen fixation by diazotrophy occurs [28], particularly in the eastern Mediterranean Sea [29]. Estimates indicate that up to 90% of the nitrogen in the eastern Mediterranean Sea may be derived from biological $N_2$ fixation in contrast with only 20% in the west [28], probably owing to the distinct gradient in $\delta^{15}N$ observed in the Mediterranean Sea from this study and others.

Isoscapes in $\delta^{13}C$ of *S. officinalis* did not show any discernable spatial trends. $\delta^{13}C$ values of marine organisms are largely a function of the local DIC, and a meta-analysis of zooplankton $\delta^{13}C$ in the North Atlantic did not find any notable patterns in the NEAO-MS [8]. The complexity of hydrodynamical, biogeochemical and ecological features in the NEAO-MS [30] may explain the lack of a clear gradient in $\delta^{13}C$ values of *S. officinalis*. Further, the heterogeneous patchiness of the phytoplankton community structure throughout pelagic waters of the Mediterranean Sea may owe to the $\delta^{13}C$ results in this study [31]. Thus, a combination of the complex physical oceanography combined with the biological community of the ecosystem may contribute to the lack of any $\delta^{13}C$ gradients observed. Two notable collection locations (Portugal–South and Aegean Sea–West) had higher $\delta^{13}C$ values in *S. officinalis* muscle tissue than other sampling locations. Specific areas of sample collection at Aegean Sea–West are unknown due to market-based sample collections; however, Portugal–South collections occurred near shore relative to other locations in this study. Near-shore waters with increased nutrient loads, higher productivity and $^{13}C$-enriched seagrass input probably contribute to higher $\delta^{13}C$ values than offshore environments [32]. In addition, $^{13}C$ inputs from benthic algae and terrestrial sources are $^{13}C$ heavy probably owing to the higher $\delta^{13}C$ values from this particular study location [32]. Future studies may want to consider increased sampling resolution as this study demonstrated local conditions probably affect the stable isotope values measured in consumer tissue.

Preservation techniques for *S. officinalis* muscle tissue did not affect stable isotopes of $\delta^{13}C$ and $\delta^{15}N$, demonstrating both methods are adequate. This finding is particularly useful for researchers collecting tissue samples of this species without immediate access to equipment such as freezers or freeze driers. Previous studies using stable isotopes on tissues of consumers including cephalopods, fishes and zooplankton have reported varying results ranging from major effects of preservation techniques to little or no impact on results [22,33,34].

Findings from this study support the use of stable isotopes as natural tracers to examine the population structure of *S. officinalis* in the NEAO-MS and add to the growing ecological toolkit of tracers available. Specifically, $\delta^{15}N$ may serve as a useful natural tracer to track movement and connectivity for marine species in the region. These findings can be extended to other marine organisms from zooplankton to apex predators to examine movement and highlight specific gradients throughout the NEAO-MS. Furthermore, *S. officinalis* are opportunistic feeders with diets comprising bony fishes, crustaceans, molluscs and polychaetes [5,35,36] and serve as prey to larger bony fishes, sharks and marine mammals [3] thereby serving as a model species for future trophic and population connectivity studies throughout the NEAO-MS.

Ethics. No ethical assessment was needed in order to carry out this research. All collections were from artisanal or commercial fishing operations, and they were caught for fish markets (not specifically for this study). All cuttlefish at markets were dead (no live specimens obtained).

Data accessibility. The data are provided in the electronic supplementary material [37].

Authors' contributions. R.J.D.W. conducted data analysis and wrote the main manuscript text; R.J.D.W. and J.R.R. designed the study; all authors assisted with sample collections and provided comments on the manuscript; all authors approve the final version and agree to be accountable for all aspects of the study.

Competing interests. We declare we have no competing interests.

Funding. This work was supported by Texas A&M University (R.J.D.W. and J.R.R.) and the McDaniel Charitable Foundation (J.R.R.). A.V.S. was funded by Fundação para a Ciência e a Tecnologia (SFRH/BPD/36100/2007). R.V. was funded by the Spanish Ministry of Science, Innovation and Universities (OCTOSET project, RTI2018-097908-B-I00, MCIU/AEI/FEDER, EU) and the 'Severo Ochoa Center of Excellence' accreditation (CEX2019-000928-S) of the Spanish Research Agency [Agencia Española de Investigación (AEI)].

Acknowledgements. Special thanks to members of the Shark Biology and Fisheries Science Lab and Fisheries Ecology Lab at Texas A&M University at Galveston for the preparation of samples for stable isotope analyses. We also appreciate the assistance of commercial and artisanal fishers that provided samples from some of the sampling locations in the Northeast Atlantic Ocean and Mediterranean Sea.

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
