## [Peer Review File · Royal Society Open Science]

Review History

RSOS-210345.R0 (Original submission)

Review form: Reviewer 1

Is the manuscript scientifically sound in its present form?

No

Are the interpretations and conclusions justified by the results?

No

Is the language acceptable?

Yes

Do you have any ethical concerns with this paper?

No

Have you any concerns about statistical analyses in this paper?

Yes

Recommendation?

Major revision is needed (please make suggestions in comments)

Comments to the Author(s)

Review Wells et al

This MS describes an investigation of the use of cuttlefish muscle tissue stable isotope values as a possible tool to identify origin.

The approach is reasonable and if successful could be valuable both as a management tool and to draw ecological inferences. A reasonable sample suite has been analysed and the data are (largely) available for re-analysis. However, I have concerns about some of the results that need to be fully addressed before any manuscript could be fully considered.

The d13C data shows that at least two sites are characterised by very high d13C values and high variance in d13C values – this implies to me that local-scale stochastic effects might have quite a big bearing on the d13C (and d15N?) values of cuttlefish tissues. If 2/11 sites showed this level of variance in one sample effort, but you don't know the cause – how confident can you be that the variance is consistent within and between sites? It seems likely that local ecosystem habitat effects are contributing to high d13C values seen (maybe seagrass fuelled food webs)? But that again raises concerns that local scale effects appear to outweigh regional effects – so unless you can be sure that there are no similar local scale effects in regions unsampled in the study, then it is dangerous to infer that you really can separate sites of origin using tissue SIA values...

I think the underlying cause likely producing these high and variable d13C values needs to be explored in much more detail, and the potential implications for the use of SIA markers needs to be spelled out.

Secondly, I don't think the data can support spatial extrapolation into isoscapes in the way presented. By extrapolating you are essentially stating that the only source of variance on tissue isotope values is the distance between sampling sites. But you know that is not true (and indeed argue that is not true when you explain high d13C values in terms of local inshore effects) – so there is a major (indeed the main) spatial effect that is not considered in the spatial model. It is very tempting to create spatial isoscape models, and we are very much in need of quality isoscapes for the Med region – but the danger of producing misleading isoscape models is very high – and I am concerned that others may be misled by the isoscape models produced here.

Abstract:

Line 32 - delta values are annotations for representing ratios (not entities in themselves) – so should be “Stable isotope compositions of carbon and nitrogen (expressed as d13C and d15N values) from the were measured (not analysed) to.....”

Line 42 and throughout refer to d13C values and d15N values (as above the delta notation is not a measurable variable in it's own right)

Introduction line 3-11 Yes, d13C values have commonly been inferred to show relatively little trophic fractionation - - however we are getting more and more evidence from single amino acid work that this is possibly a big oversimplification - so perhaps the point here is that among-individual physiological effects on tissue d13C values are on balance small in scale compared to among-region effects on d13C baseline values...

I think this MS would benefit from much more background on existing datasets exploring differences in d13C and d15N values within the NEAO-Med system - there are quite a lot of discrete studies, although few (no as far as I'm aware) large scale data compilations. The MS would have much more solidity if it were anchored on a more thorough characterisation of consumer isotope values expected in different regions based on (admittedly patchy and limited) literature data available

Methods p5 line 42: CN ratios between 3.5 and 4 may well be influenced by lipid contents - can you plot CN ratios against d13C values? (CN ratios not given in the supp data).

Page 7 lines 1-34. Could debate whether these are the most appropriate statistical methods. I'm not sure that the design is really appropriate for null hypothesis significance testing - there isn't a true null hypothesis being tested (you already had an expectation that there will be regional effect on SI values) - the general trend away from NHST would probably argue at least that setting store on $\alpha=0.05$ as an arbitrary significance value should be dropped.

e.g. <https://www.amstat.org/asa/files/pdfs/P-ValueStatement.pdf>

<https://www.tandfonline.com/doi/full/10.1080/00031305.2019.1583913>

(ANOVA etc) tests whether means vary among regions while the aim of the QDFA is to test whether individual cases can be discriminated among regions. That distinction is important. But it's also important to note that here you don't have independent training and testing datasets for discriminant analysis - so the output will be a 'best possible' assessment of discriminatory ability.

Results -

P6 line 52 - here is an example where I would again question p values as an arbiter of effect. Is it possible that there is a preservation effect on d13C values given the relatively small sample size and p value potentially implying non-random partitioning of variance between treatments, but not reaching the arbitrary p value standard??

I am rather concerned about the absolute d13C values and associated variances recorded in the Portugal south and Aegean west (J) samples. D13C values $>-13\%$ are fairly unusual for muscle tissue from marine organisms (and imply food web baseline values very much higher than expected in these regions). These two sites are also linked to higher standard deviations of d13C values - which further implies that at the least they are not truly representative of a stable regional isotopic composition - whether that is because of localised point variance in food web d13C values or some form of sample treatment issue.

Could seagrass-influenced food webs be contributing here? The problem raised, then, is that if by chance 2/11 discrete sites show so much variation due to potentially very localised habitat effects - how can you be confident that the differences observed in this sampling effort would be replicated in an independent sampling at slightly different sites / times of year / years?

I am not entirely sold on the idea of extrapolating isoscapes based on discrete samples with no additional spatial rules - i.e. a simple distance kriging model as applied here. Even without the highly variable Portugal S and Aegean W samples, by extrapolating you are suggesting that you can predict into un-sampled areas based on the samples gathered. I'm not convinced that that is advisable - particularly without having a stronger mechanistic explanation justifying extrapolation into unsampled areas. I agree that isoscapes from the Med are badly needed, but I would be very very cautious about providing spatial models based on limited regionally discrete samples. The high variances seen in 2/11 sites makes me even more nervous about doing this.

Review form: Reviewer 2

Is the manuscript scientifically sound in its present form?

Yes

Are the interpretations and conclusions justified by the results?

Yes

Is the language acceptable?

Yes

Do you have any ethical concerns with this paper?

No

Have you any concerns about statistical analyses in this paper?

No

Recommendation?

Accept with minor revision (please list in comments)

Comments to the Author(s)

Review of RSOS-210345 Regional patterns of $\delta^{13}\text{C}$ and $\delta^{15}\text{N}$ for European common cuttlefish (*Sepia officinalis*) throughout the Northeast Atlantic Ocean and Mediterranean Sea

The author's assessed $\delta^{13}\text{C}$ and $\delta^{15}\text{N}$ isotope values in the tissue of European common cuttlefish collected from 11 sites in the Northeast Atlantic Ocean and the Mediterranean Sea. This is a well written and straightforward manuscript and the dataset and analyses are adequate to meet the authors' stated objective: to examine patterns in stable isotopes from common cuttlefish collected from the Northeast Atlantic Ocean and Mediterranean Sea. The data presented in this manuscript should make a useful contribution to the field. I have a few specific comments (listed below) for the authors to consider. All of them could be easily addressed in a revision. My most substantive comment relates to the authors analysis of preserved and unpreserved samples (comment 4 below). Specifically, I recommend that they identify the preservation method of each sample in the electronic supplementary material.

1. Page 3 line 11. Meaning is unclear, I would suggest a revision. Perhaps: " $\delta^{13}\text{C}$ values of consumers are a product of the composition of primary producers in the diet and is...."
2. Page 3 line 26. "...high nutrient (nitrate)..." Are the authors specifically referring to nitrate pollution? If so, please state parenthetically (as is the pattern for the other sources in this section).
3. Page 5 line 6. Meaning of "small sizes" is unclear. Do you mean that care was taken to ensure that only a small proportion of the sample from a given region was acquired from local fishers? If so, please clarify (perhaps with similar wording). Perhaps including data on the percentage of samples acquired from local fishers or adding the source of cuttlefish to the existing supplementary data (i.e. add a column) would be helpful.
4. Page 6 results and supplemental information. The authors acknowledge that some of their samples were freeze dried while others were preserved in ethanol. The authors performed an analysis that suggests there were no significant effects of ethanol preservation on isotope values. I would recommend that the authors identify which samples were preserved in ethanol and which were freeze dried in the electronic supplementary material (i.e. add a column to the

dataset). This will allow the reader to replicate the authors' analysis. It also might be useful for the authors to provide a simple figure or table in the supplementary material that would give the reader insight into the mean and variance associated with these samples. I think "seeing" the data would enhance confidence in this result and could even be useful for other authors working with similar datasets (i.e. preserved and unpreserved samples).

5. Page 8, line 25. Is it possible to define "limited dispersal"? If a quantitative estimate exists it might be helpful to provide it here (perhaps parenthetically).

6. Page 11, line 18. The authors end the introduction by making a nice point that this study could enhance the ecological toolkit of "tracers" available to scientists working with cuttlefish in the region. It might be nice to revisit this point at the end. Specifically, the authors found that stable isotopes of C and N could be useful tracers in and of themselves in some contexts – but in combination with other "tools" or tracers, high resolution (or even higher resolution) tracer studies may be possible.

Decision letter (RSOS-210345.R0)

Dear Dr Wells

The Editors assigned to your paper RSOS-210345 "Regional patterns of $\delta^{13}\text{C}$ and $\delta^{15}\text{N}$ for European common cuttlefish (*Sepia officinalis*) throughout the Northeast Atlantic Ocean and Mediterranean Sea" have now received comments from reviewers and would like you to revise the paper in accordance with the reviewer comments and any comments from the Editors. Please note this decision does not guarantee eventual acceptance.

Please submit your revised manuscript and required files (see below) no later than 21 days from today's (ie 01-Jul-2021) date. Note: the ScholarOne system will 'lock' if submission of the revision is attempted 21 or more days after the deadline. If you do not think you will be able to meet this deadline please contact the editorial office immediately.

Please note article processing charges apply to papers accepted for publication in Royal Society Open Science (<https://royalsocietypublishing.org/rsos/charges>). Charges will also apply to papers transferred to the journal from other Royal Society Publishing journals, as well as papers submitted as part of our collaboration with the Royal Society of Chemistry

(<https://royalsocietypublishing.org/rsos/chemistry>). Fee waivers are available but must be requested when you submit your revision (<https://royalsocietypublishing.org/rsos/waivers>).

on behalf of Dr Melita Samoilys (Associate Editor) and Pete Smith (Subject Editor)
openscience@royalsociety.org

Editor Comments to Author:

Based on the two referee reports we've received on this submission, this paper requires a 'Major Revision'. This decision will give the authors time to address the referees' comments.

Reviewer comments to Author:

Reviewer: 1

Comments to the Author(s)

Review Wells et al

This MS describes an investigation of the use of cuttlefish muscle tissue stable isotope values as a possible tool to identify origin.

The approach is reasonable and if successful could be valuable both as a management tool and to draw ecological inferences. A reasonable sample suite has been analysed and the data are (largely) available for re-analysis. However, I have concerns about some of the results that need to be fully addressed before any manuscript could be fully considered.

The $\delta^{13}\text{C}$ data shows that at least two sites are characterised by very high $\delta^{13}\text{C}$ values and high variance in $\delta^{13}\text{C}$ values – this implies to me that local-scale stochastic effects might have quite a big bearing on the $\delta^{13}\text{C}$ (and $\delta^{15}\text{N}$?) values of cuttlefish tissues. If 2/11 sites showed this level of variance in one sample effort, but you don't know the cause – how confident can you be that the variance is consistent within and between sites? It seems likely that local ecosystem habitat effects are contributing to high $\delta^{13}\text{C}$ values seen (maybe seagrass fuelled food webs)? But that again raises concerns that local scale effects appear to outweigh regional effects – so unless you can be sure that there are no similar local scale effects in regions unsampled in the study, then it is dangerous to infer that you really can separate sites of origin using tissue SIA values...

I think the underlying cause likely producing these high and variable $\delta^{13}\text{C}$ values needs to be explored in much more detail, and the potential implications for the use of SIA markers needs to be spelled out.

Secondly, I don't think the data can support spatial extrapolation into isoscapes in the way presented. By extrapolating you are essentially stating that the only source of variance on tissue isotope values is the distance between sampling sites. But you know that is not true (and indeed argue that is not true when you explain high $\delta^{13}\text{C}$ values in terms of local inshore effects) – so there is a major (indeed the main) spatial effect that is not considered in the spatial model. It is very tempting to create spatial isoscape models, and we are very much in need of quality isoscapes for the Med region – but the danger of producing misleading isoscape models is very high – and I am concerned that others may be misled by the isoscape models produced here.

Abstract:

Line 32 - delta values are annotations for representing ratios (not entities in themselves) - so should be "Stable isotope compositions of carbon and nitrogen (expressed as d13C and d15N values) from the were measured (not analysed) to....."

Line 42 and throughout refer to d13C values and d15N values (as above the delta notation is not a measurable variable in it's own right)

Introduction line 3-11 Yes, d13C values have commonly been inferred to show relatively little trophic fractionation - - however we are getting more and more evidence from single amino acid work that this is possibly a big oversimplification - so perhaps the point here is that among-individual physiological effects on tissue d13C values are on balance small in scale compared to among-region effects on d13C baseline values...

I think this MS would benefit from much more background on existing datasets exploring differences in d13C and d15N values within the NEAO-Med system - there are quite a lot of discrete studies, although few (no as far as I'm aware) large scale data compilations. The MS would have much more solidity if it were anchored on a more thorough characterisation of consumer isotope values expected in different regions based on (admittedly patchy and limited) literature data available

Methods p5 line 42: CN ratios between 3.5 and 4 may well be influenced by lipid contents - can you plot CN ratios against d13C values? (CN ratios not given in the supp data).

Page 7 lines 1-34. Could debate whether these are the most appropriate statistical methods. I'm not sure that the design is really appropriate for null hypothesis significance testing - there isn't a true null hypothesis being tested (you already had an expectation that there will be regional effect on SI values) - the general trend away from NHST would probably argue at least that setting store on $\alpha=0.05$ as an arbitrary significance value should be dropped.

e.g. <https://www.amstat.org/asa/files/pdfs/P-ValueStatement.pdf>

<https://www.tandfonline.com/doi/full/10.1080/00031305.2019.1583913>

(ANOVA etc) tests whether means vary among regions while the aim of the QDFA is to test whether individual cases can be discriminated among regions. That distinction is important. But it's also important to note that here you don't have independent training and testing datasets for discriminant analysis - so the output will be a 'best possible' assessment of discriminatory ability.

Results -

P6 line 52 - here is an example where I would again question p values as an arbiter of effect. Is it possible that there is a preservation effect on d13C values given the relatively small sample size and p value potentially implying non-random partitioning of variance between treatments, but not reaching the arbitrary p value standard??

I am rather concerned about the absolute d13C values and associated variances recorded in the Portugal south and Aegean west (J) samples. D13C values $>-13\%$ are fairly unusual for muscle tissue from marine organisms (and imply food web baseline values very much higher than expected in these regions). These two sites are also linked to higher standard deviations of d13C values - which further implies that at the least they are not truly representative of a stable regional isotopic composition - whether that is because of localised point variance in food web d13C values or some form of sample treatment issue.

Could seagrass-influenced food webs be contributing here? The problem raised, then, is that if by chance 2/11 discrete sites show so much variation due to potentially very localised habitat effects – how can you be confident that the differences observed in this sampling effort would be replicated in an independent sampling at slightly different sites / times of year / years?

I am not entirely sold on the idea of extrapolating isoscapes based on discrete samples with no additional spatial rules – i.e. a simple distance kriging model as applied here. Even without the highly variable Portugal S and Aegean W samples, by extrapolating you are suggesting that you can predict into un-sampled areas based on the samples gathered. I'm not convinced that that is advisable – particularly without having a stronger mechanistic explanation justifying extrapolation into unsampled areas. I agree that isoscapes from the Med are badly needed, but I would be very very cautious about providing spatial models based on limited regionally discrete samples. The high variances seen in 2/11 sites makes me even more nervous about doing this.

Reviewer: 2

Comments to the Author(s)

Review of RSOS-210345 Regional patterns of $\delta^{13}\text{C}$ and $\delta^{15}\text{N}$ for European common cuttlefish (*Sepia officinalis*) throughout the Northeast Atlantic Ocean and Mediterranean Sea

The author's assessed $\delta^{13}\text{C}$ and $\delta^{15}\text{N}$ isotope values in the tissue of European common cuttlefish collected from 11 sites in the Northeast Atlantic Ocean and the Mediterranean Sea. This is a well written and straightforward manuscript and the dataset and analyses are adequate to meet the authors' stated objective: to examine patterns in stable isotopes from common cuttlefish collected from the Northeast Atlantic Ocean and Mediterranean Sea. The data presented in this manuscript should make a useful contribution to the field. I have a few specific comments (listed below) for the authors to consider. All of them could be easily addressed in a revision. My most substantive comment relates to the authors analysis of preserved and unpreserved samples (comment 4 below). Specifically, I recommend that they identify the preservation method of each sample in the electronic supplementary material.

1. Page 3 line 11. Meaning is unclear, I would suggest a revision. Perhaps: " $\delta^{13}\text{C}$ values of consumers are a product of the composition of primary producers in the diet and is...."
2. Page 3 line 26. "...high nutrient (nitrate)..." Are the authors specifically referring to nitrate pollution? If so, please state parenthetically (as is the pattern for the other sources in this section).
3. Page 5 line 6. Meaning of "small sizes" is unclear. Do you mean that care was taken to ensure that only a small proportion of the sample from a given region was acquired from local fishers? If so, please clarify (perhaps with similar wording). Perhaps including data on the percentage of samples acquired from local fishers or adding the source of cuttlefish to the existing supplementary data (i.e. add a column) would be helpful.
4. Page 6 results and supplemental information. The authors acknowledge that some of their samples were freeze dried while others were preserved in ethanol. The authors performed an analysis that suggests there were no significant effects of ethanol preservation on isotope values. I would recommend that the authors identify which samples were preserved in ethanol and which were freeze dried in the electronic supplementary material (i.e. add a column to the dataset). This will allow the reader to replicate the authors' analysis. It also might be useful for the authors to provide a simple figure or table in the supplementary material that would give the reader insight into the mean and variance associated with these samples. I think "seeing" the

data would enhance confidence in this result and could even be useful for other authors working with similar datasets (i.e. preserved and unpreserved samples).

5. Page 8, line 25. Is it possible to define “limited dispersal”? If a quantitative estimate exists it might be helpful to provide it here (perhaps parenthetically).

6. Page 11, line 18. The authors end the introduction by making a nice point that this study could enhance the ecological toolkit of “tracers” available to scientists working with cuttlefish in the region. It might be nice to revisit this point at the end. Specifically, the authors found that stable isotopes of C and N could be useful tracers in and of themselves in some contexts – but in combination with other “tools” or tracers, high resolution (or even higher resolution) tracer studies may be possible.

===PREPARING YOUR MANUSCRIPT===

===PREPARING YOUR REVISION IN SCHOLARONE===

Author's Response to Decision Letter for (RSOS-210345.R0)

See Appendix A.

RSOS-210345.R1 (Revision)

Review form: Reviewer 2

Is the manuscript scientifically sound in its present form?

Yes

Are the interpretations and conclusions justified by the results?

Yes

Is the language acceptable?

Yes

Do you have any ethical concerns with this paper?

No

Have you any concerns about statistical analyses in this paper?

No

Recommendation?

Accept with minor revision (please list in comments)

Comments to the Author(s)

Review of Manuscript ID RSOS-210345.R1, entitled "Regional patterns of $\delta^{13}\text{C}$ and $\delta^{15}\text{N}$ for European common cuttlefish (*Sepia officinalis*) throughout the Northeast Atlantic Ocean and Mediterranean Sea."

I reviewed the original manuscript. The authors addressed each comment from the two original reviewers and have made appropriate revisions to the manuscript. I am comfortable recommending the manuscript for acceptance.

I only have two minor comments related to the supplemental table:

1. The authors added data from their preservation study to the supplemental table, it might be helpful to reference these data in the text when appropriate (e.g. page 7, line 8).
2. In addition, I recommend adding a table legend to help those readers that download the supplemental data make sense of the columns (which columns/ data were used in the primary analyses, which were used in the preservation comparison, etc). Similarly, the column headings "C" and "N" should be updated to better reflect the fact that these columns contain data on C and N isotope values (i.e. $\delta^{13}\text{C}$).

Decision letter (RSOS-210345.R1)

Dear Dr Wells

On behalf of the Editors, we are pleased to inform you that your Manuscript RSOS-210345.R1 "Regional patterns of δ 13C and δ 15N for European common cuttlefish (*Sepia officinalis*) throughout the Northeast Atlantic Ocean and Mediterranean Sea" has been accepted for publication in Royal Society Open Science subject to minor revision in accordance with the referees' reports. Please find the referees' comments along with any feedback from the Editors below my signature.

Please submit your revised manuscript and required files (see below) no later than 7 days from today's (ie 09-Aug-2021) date. Note: the ScholarOne system will 'lock' if submission of the revision is attempted 7 or more days after the deadline. If you do not think you will be able to meet this deadline please contact the editorial office immediately.

on behalf of Pete Smith (Subject Editor)
openscience@royalsociety.org

Associate Editor Comments to Author:

Associate Editor: 1

Comments to the Author:

Thank you for this revision. As you'll see, the reviewer only has a couple of minor comments/queries remaining.

Reviewer comments to Author:

Reviewer: 2

Comments to the Author(s)

Review of Manuscript ID RSOS-210345.R1, entitled "Regional patterns of δ 13C and δ 15N for European common cuttlefish (*Sepia officinalis*) throughout the Northeast Atlantic Ocean and Mediterranean Sea."

I reviewed the original manuscript. The authors addressed each comment from the two original reviewers and have made appropriate revisions to the manuscript. I am comfortable recommending the manuscript for acceptance.

I only have two minor comments related to the supplemental table:

1. The authors added data from their preservation study to the supplemental table, it might be helpful to reference these data in the text when appropriate (e.g. page 7, line 8).
2. In addition, I recommend adding a table legend to help those readers that download the supplemental data make sense of the columns (which columns/ data were used in the primary analyses, which were used in the preservation comparison, etc). Similarly, the column headings "C" and "N" should be updated to better reflect the fact that these columns contain data on C and N isotope values (i.e. deltaC13).

===PREPARING YOUR MANUSCRIPT===

===PREPARING YOUR REVISION IN SCHOLARONE===

Author's Response to Decision Letter for (RSOS-210345.R1)

See Appendix B.

Decision letter (RSOS-210345.R2)

Dear Dr Wells,

I am pleased to inform you that your manuscript entitled "Regional patterns of $\delta^{13}\text{C}$ and $\delta^{15}\text{N}$ for European common cuttlefish (*Sepia officinalis*) throughout the Northeast Atlantic Ocean and Mediterranean Sea" is now accepted for publication in Royal Society Open Science.

on behalf of Prof Pete Smith (Subject Editor)
openscience@royalsociety.org

Follow Royal Society Publishing on Twitter: @RSocPublishing
Follow Royal Society Publishing on Facebook:
<https://www.facebook.com/RoyalSocietyPublishing.FanPage/>

Read Royal Society Publishing's blog:
<https://royalsociety.org/blog/blogsearchpage/?category=Publishing>

Appendix A

DEPARTMENT OF MARINE BIOLOGY

July 7, 2021

Dear Royal Society Open Science,

Thank you for advising us on the favorable reviews of our manuscript entitled “Regional patterns of $\delta^{13}\text{C}$ and $\delta^{15}\text{N}$ for European common cuttlefish (*Sepia officinalis*) throughout the Northeast Atlantic Ocean and Mediterranean Sea” for review and publication in Royal Society Open Science. We have carefully considered all of the referees’ comments and incorporated these into the revised manuscript. Please see the specific edits and responses made to each comment below.

We thank the reviewers for the valuable comments and consequently feel that the manuscript has improved. Please do not hesitate to contact me for any clarification needed during the review process.

Thank you for consideration of this manuscript.

Sincerely,

David Wells

Dr. R. J. David Wells
Texas A&M University at Galveston
Department of Marine Biology
1001 Texas Clipper Rd.
Galveston, TX 77553
Tel: 409-740-4989
wellsr@tamug.edu

In this letter, we have replied (*italicized blue text*) to the specific comments from each reviewer:

P.O. Box 1675 Galveston, TX 77553
Office : 409-740-4529

Reviewer #1:

Review Wells et al

This MS describes an investigation of the use of cuttlefish muscle tissue stable isotope values as a possible tool to identify origin.

The approach is reasonable and if successful could be valuable both as a management tool and to draw ecological inferences. A reasonable sample suite has been analysed and the data are (largely) available for re-analysis. However, I have concerns about some of the results that need to be fully addressed before any manuscript could be fully considered.

We appreciate the positive review and have incorporated the requested revisions in response to reviewer #1.

The d13C data shows that at least two sites are characterised by very high d13C values and high variance in d13C values – this implies to me that local-scale stochastic effects might have quite a big bearing on the d13C (and d15N?) values of cuttlefish tissues. If 2/11 sites showed this level of variance in one sample effort, but you don't know the cause – how confident can you be that the variance is consistent within and between sites? It seems likely that local ecosystem habitat effects are contributing to high d13C values seen (maybe seagrass fuelled food webs)? But that again raises concerns that local scale effects appear to outweigh regional effects – so unless you can be sure that there are no similar local scale effects in regions unsampled in the study, then it is dangerous to infer that you really can separate sites of origin using tissue SIA values...

We agree with the reviewer and have addressed the issue through revised text in the discussion. We have also created a new figure 2 that relates to this same issue of local scale variability and the request not to extrapolate our isoscapes beyond the scope of this study.

Secondly, I don't think the data can support spatial extrapolation into isoscapes in the way presented. By extrapolating you are essentially stating that the only source of variance on tissue isotope values is the distance between sampling sites. But you know that is not true (and indeed argue that is not true when you explain high d13C values in terms of local inshore effects) – so there is a major (indeed the main) spatial effect that is not considered in the spatial model. It is very tempting to create spatial isoscape models, and we are very much in need of quality isoscapes for the Med region – but the danger of producing misleading isoscape models is very high – and I am concerned that others may be misled by the isoscape models produced here.

We agree and have therefore modified the isoscape figures (Figure 2) to support the measured values in the study without extrapolating beyond our collection locations. The revised figure also gives readers a measure of both mean and individual variability within each sampling location.

Abstract:

Line 32 - delta values are annotations for representing ratios (not entities in themselves) – so should be “Stable isotope compositions of carbon and nitrogen (expressed as d13C and d15N values) from the were measured (not analysed) to.....”

Changed as suggested

Line 42 and throughout refer to d13C values and d15N values (as above the delta notation is not a measurable variable in it's own right)

Changed as suggested

Methods p5 line 42: CN ratios between 3.5 and 4 may well be influenced by lipid contents – can you plot CN ratios against d13C values? (CN ratios not given in the supp data).

Mean C:N was 3.21 ± 0.13 SD (ranging from 2.87 to 3.75) confirming samples were not influenced by lipid contents (Post et al. 2007). We have added text in the manuscript for support.

Page 7 lines 1-34. Could debate whether these are the most appropriate statistical methods. I'm not sure that the design is really appropriate for null hypothesis significance testing – there isn't a true null hypothesis being tested (you already had an expectation that there will be regional effect on SI values) – the general trend away from NHST would probably argue at least that setting store on $\alpha=0.05$ as an arbitrary significance value should be dropped.

e.g. <https://www.amstat.org/asa/files/pdfs/P-ValueStatement.pdf>

<https://www.tandfonline.com/doi/full/10.1080/00031305.2019.1583913>

(ANOVA etc) tests whether means vary among regions while the aim of the QDFA is to test whether individual cases can be discriminated among regions. That distinction is important. But it's also important to note that here you don't have independent training and testing datasets for discriminant analysis - so the output will be a 'best possible' assessment of discriminatory ability.

Thank you for the suggestion and input here. We agree the QDFA is indeed the "best possible" classification and have added this text in revised manuscript. Because our objectives of the study were to test if differences in C and N exist among sampling locations we prefer to remain with the standard statistical significance alpha level of 0.05 (p-value).

Results –

P6 line 52 – here is an example where I would again question p values as an arbiter of effect. Is it possible that there is a preservation effect on d13C values given the relatively small sample size and p value potentially implying non-random partitioning of variance between treatments, but not reaching the arbitrary p value standard??

We appreciate the comment. A sample size of 20 to compare preservation techniques should encompass the variability we could capture in this study as this sample size equaled or exceeded all but two regions total sample sizes. Equality of variance between treatments was not an issue.

I am rather concerned about the absolute d13C values and associated variances recorded in the Portugal south and Aegean west (J) samples. D13C values $>-13\text{‰}$ are fairly unusual for muscle tissue from marine organisms (and imply food web baseline values very much higher than expected in these regions). These two sites are also linked to higher standard deviations of d13C values – which further implies that at the least they are not truly representative of a stable regional isotopic composition – whether that is because of localised point variance in food web d13C values or some form of sample treatment issue.

Could seagrass-influenced food webs be contributing here? The problem raised, then, is that if

by chance 2/11 discrete sites show so much variation due to potentially very localised habitat effects – how can you be confident that the differences observed in this sampling effort would be replicated in an independent sampling at slightly different sites / times of year / years?

We appreciate the reviewers input here and agree the carbon values are unique in these two regions. Collection locations at the Portugal south location was nearshore relative to the other collection locations and likely a function of localized conditions. We have added additional text in the manuscript discussing this issue, including the potential influence of seagrass enriched values. We do not have any collection information for the market-based samples from the Aegean west location so were limited in speculating the mechanisms driving the enriched carbon results. However, we still have text devoted to possible mechanisms related to this comment.

I am not entirely sold on the idea of extrapolating isoscapes based on discrete samples with no additional spatial rules – i.e. a simple distance kriging model as applied here. Even without the highly variable Portugal S and Aegean W samples, by extrapolating you are suggesting that you can predict into un-sampled areas based on the samples gathered. I'm not convinced that that is advisable - particularly without having a stronger mechanistic explanation justifying extrapolation into unsampled areas. I agree that isoscapes from the Med are badly needed, but I would be very very cautious about providing spatial models based on limited regionally discrete samples. The high variances seen in 2/11 sites makes me even more nervous about doing this.

We agree with the reviewer and have therefore modified the isoscape plots (Figure 2) to show the trends without extrapolating outside of the regions that were not sampled. The individual C & N values can also be seen in the revised figure while still showing the mean values by sampling location.

Reviewer: 2

Review of RSOS-210345 Regional patterns of $\delta^{13}\text{C}$ and $\delta^{15}\text{N}$ for European common cuttlefish (*Sepia officinalis*) throughout the Northeast Atlantic Ocean and Mediterranean Sea

The author's assessed $\delta^{13}\text{C}$ and $\delta^{15}\text{N}$ isotope values in the tissue of European common cuttlefish collected from 11 sites in the Northeast Atlantic Ocean and the Mediterranean Sea. This is a well written and straightforward manuscript and the dataset and analyses are adequate to meet the authors' stated objective: to examine patterns in stable isotopes from common cuttlefish collected from the Northeast Atlantic Ocean and Mediterranean Sea. The data presented in this manuscript should make a useful contribution to the field. I have a few specific comments (listed below) for the authors to consider. All of them could be easily addressed in a revision. My most substantive comment relates to the authors analysis of preserved and unpreserved samples (comment 4 below). Specifically, I recommend that they identify the preservation method of each sample in the electronic supplementary material.

We appreciate the positive review and have incorporated the requested revisions in response to reviewer #2. Specifically, we added the preservation method of each sample in the supplementary material as suggested.

1. Page 3 line 11. Meaning is unclear, I would suggest a revision. Perhaps: " $\delta^{13}\text{C}$ values of

consumers are a product of the composition of primary producers in the diet and is....”
Revised as suggested.

2. Page 3 line 26. “...high nutrient (nitrate)...” Are the authors specifically referring to nitrate pollution? If so, please state parenthetically (as is the pattern for the other sources in this section).

The goal is to contrast high vs low and new vs regenerated nitrogen (not necessarily nitrate pollution).

3. Page 5 line 6. Meaning of “small sizes” is unclear. Do you mean that care was taken to ensure that only a small proportion of the sample from a given region was acquired from local fishers? If so, please clarify (perhaps with similar wording). Perhaps including data on the percentage of samples acquired from local fishers or adding the source of cuttlefish to the existing supplementary data (i.e. add a column) would be helpful.

Text has been modified. Small “body” sizes to ensure we were not collecting large individuals that may have been at large for a long period of time and moved in from another location/region.

4. Page 6 results and supplemental information. The authors acknowledge that some of their samples were freeze dried while others were preserved in ethanol. The authors performed an analysis that suggests there were no significant effects of ethanol preservation on isotope values. I would recommend that the authors identify which samples were preserved in ethanol and which were freeze dried in the electronic supplementary material (i.e. add a column to the dataset). This will allow the reader to replicate the authors’ analysis. It also might be useful for the authors to provide a simple figure or table in the supplementary material that would give the reader insight into the mean and variance associated with these samples. I think “seeing” the data would enhance confidence in this result and could even be useful for other authors working with similar datasets (i.e. preserved and unpreserved samples).

We added the preservation method of each sample in supplementary material, as suggested.

5. Page 8, line 25. Is it possible to define “limited dispersal”? If a quantitative estimate exists it might be helpful to provide it here (perhaps parenthetically).

It would be misleading to quantify this term. We prefer to keep it general rather than misinforming a reader with a quantitative estimate since we did not actually measure movement and dispersal in this study.

6. Page 11, line 18. The authors end the introduction by making a nice point that this study could enhance the ecological toolkit of “tracers” available to scientists working with cuttlefish in the region. It might be nice to revisit this point at the end. Specifically, the authors found that stable isotopes of C and N could be useful tracers in and of themselves in some contexts – but in combination with other “tools” or tracers, high resolution (or even higher resolution) tracer studies may be possible.

Added text, as suggested.

Appendix B

DEPARTMENT OF MARINE BIOLOGY

August 10, 2021

Dear Royal Society Open Science,

Thank you for advising us on the acceptance of our manuscript entitled “Regional patterns of $\delta^{13}\text{C}$ and $\delta^{15}\text{N}$ for European common cuttlefish (*Sepia officinalis*) throughout the Northeast Atlantic Ocean and Mediterranean Sea” for publication in Royal Society Open Science. We have made the two minor changes requested to the manuscript. Please see the specific edits and responses made to each comment below.

Please do not hesitate to contact me for any clarification needed.

Sincerely,

David Wells

Dr. R. J. David Wells
Texas A&M University at Galveston
Department of Marine Biology
1001 Texas Clipper Rd.
Galveston, TX 77553
Tel: 409-740-4989
wellsr@tamug.edu

We have replied (*italicized blue text*) to the specific comments below:

The authors added data from their preservation study to the supplemental table, it might be helpful to reference these data in the text when appropriate (e.g. page 7, line 8).

Changed and added text/reference as suggested.

In addition, I recommend adding a table legend to help those readers that download the supplemental data make sense of the columns (which columns/data were used in the primary analyses, which were used in the preservation comparison, etc). Similarly, the column headings “C” and “N” should be updated to better reflect the fact that these columns contain data on C and N isotope values (i.e. deltaC13).

Changed supplemental data as suggested.

P.O. Box 1675 Galveston, TX 77553
Office : 409-740-4529